# Serious Games for Developing Social Skills in Children and Adolescents with Autism Spectrum Disorder: A Systematic Review

**DOI:** 10.3390/healthcare12050508

**Published:** 2024-02-20

**Authors:** Tânia Carneiro, António Carvalho, Sónia Frota, Marisa G. Filipe

**Affiliations:** 1Center of Linguistics, School of Arts and Humanities, University of Lisbon, 1600-214 Lisboa, Portugal; tania.s.carneiro@edu.ulisboa.pt (T.C.); sfrota@edu.ulisboa.pt (S.F.); 2Faculty of Psychology, Education, and Sports, Lusófona University, 4000-098 Porto, Portugal

**Keywords:** systematic review, autism spectrum disorder, serious games, social skills

## Abstract

Serious games represent a promising avenue for intervention with children diagnosed with autism spectrum disorder, a neurodevelopmental disorder marked by persistent challenges in social communication and the presence of restricted, repetitive behaviors. Despite this potential, comprehensive reviews on this subject are scarce. This systematic review aims to evaluate the effectiveness of serious games and their specific characteristics in enhancing social skills among children and adolescents with autism. Employing PICO strategies and adhering to PRISMA guidelines, we screened 149 studies initially identified through PubMed and EBSCOhost databases. Nine studies met inclusion criteria and found a positive influence of serious games on social skills and related domains, encompassing emotion recognition/encoding/decoding, emotional regulation, eye gaze, joint attention, and behavioral skills. Nevertheless, despite these promising results, the limited available evidence underscores the need for rigorous study designs to consolidate findings and integrate evidence-based intervention strategies.

## 1. Introduction

Autism spectrum disorder (ASD) is a neurodevelopmental disorder characterized by persistent impairments in social communication and restricted, repetitive patterns of behavior [1]. One defining aspect of this disorder is the difficulty with social interactions. Scholars, such as Baron-Cohen [2], have proposed that impairments in social skills span a broad spectrum. At one end, some children tend toward solitude, avoiding interactions with others. Conversely, at the other end, some actively seek social engagement but lack the know-how to initiate and sustain communication, leading to frequently inappropriate social interactions. Research has found social impairments in ASD in the domains of social cognition, social perception, and social attention [3]. Indeed, individuals with autism face difficulties in utilizing and comprehending fundamental social cues, including eye contact, facial expressions, body language, and variations in tone of voice during communication [2]. Furthermore, research indicates that children with ASD struggle with interpreting the thoughts and feelings of others, initiating interactions, maintaining conversations, responding to requests, and participating in cooperative activities [4,5,6].

The social communicative challenges linked to ASD underscore the critical importance of targeting social skills in interventions, as Wolstencroft et al. [7] highlighted. In the past decade, there have been significant advancements in social skills interventions [8], including evidence-based approaches such as real-world practice, video-based modeling, peer-assisted interventions, group training [9], priming, self-management, written scripts, social stories, and pivotal response training [10]. Despite the increasing number of empirical studies and reviews on interventions for children with ASD, no widely accepted practices have emerged for this particular group [8]. Nevertheless, a critical factor that significantly contributes to enhancing social skills and promoting voluntary social engagement is the learner’s interest. Therefore, it is crucial for social skills programs to actively enhance an individual’s focus and enthusiasm toward social stimuli. This involves not only addressing specific tasks but also aligning these tasks with the individual’s intrinsic interests. Considering these insights, the role of technology gains significance, particularly for individuals with ASD who often find comfort in predictable environments free from social demands [9]. Research has shown that children with ASD often find enjoyment in computerized interventions and demonstrate notable learning advancements through diverse technology-driven training, such as video-based modeling, computer-assisted teaching, video games, and virtual environments [11,12]. Therefore, digital technologies for training social skills have emerged as a promising approach, offering tailored interventions that can effectively engage individuals with ASD, aligning with their distinctive interests and learning preferences.

Computer-based instruction is widely employed in special education, providing a valuable tool for educators working with children diagnosed with ASD [13,14,15]. Several review articles have explored the application of computer-based instruction for intervening in various skills relevant to children with ASD and have demonstrated that serious games hold promise as tools to address and enhance different skills in this population [16,17,18,19]. Serious games are games where entertainment, fun, or enjoyment are not the primary focus [20]. The term “serious” denotes their function in communicating a specific message or transferring knowledge, skills, or other content to the player [21]. With their educational focus, serious games are explicitly designed to enhance users’ skills rather than merely entertainment [22]. Additionally, these games embody game-based technology addressing significant real-world issues in domains such as education and healthcare [23].

Several reviews have investigated the use of serious games as an adjunct intervention for individuals with autism ASD [24,25,26,27]. Notably, these reviews have highlighted the development of various serious games tailored to support children with ASD in acquiring diverse skills, including communication, learning, social behavior, and motor abilities through different approaches [13,25,27]. The evidence supporting the use of serious games in ASD interventions has shown effectiveness, and improvements have been particularly evident in social skills, for instance, as indicated by the recent systematic review conducted by Silva et al. [28] that compares the use of serious games and entertainment games in interventions targeting ASD symptoms [28].

Despite the few reviews available, a significant gap exists in the depth of exploration concerning the specific impact of serious games on the social skills of individuals with ASD. While prior reviews have examined the impact of serious games on social skills, these studies did not explore specific social skills (e.g., joint attention, eye gaze), instead focusing on general social skills. This lack of specificity poses a challenge as it hinders a detailed understanding of the nuanced effects of serious games on distinct aspects of social skills. Additionally, prior reviews did not examine the characteristics of games that could improve the different skills more efficiently.

Research has emphasized the importance of integrating several key characteristics into serious games, as well as combining design and learning strategies with appropriate instructional methods and pedagogical models. This integration is a critical step in enriching the learning experience within the gaming environment [29]. Serious games need to be well-designed, grounded on sound theory, and engaging for the user [13]. This entails that the success of even a meticulously designed game hinges on its effective implementation.

### Present Study

As individuals diagnosed with ASD encounter distinctive challenges in acquiring and developing social skills [30], the main goal of the current study is to systematically review the evidence on the application of serious games as a tool to enhance these specific skills in children and adolescents with ASD. In this exploration, we aim to delve into the impacts of serious games on distinct social competencies and their characteristics. Moreover, this review seeks to streamline the selection process for serious games used in interventions.

It is crucial to note that research has presented various definitions of social skills, with the term often used interchangeably with concepts like social competence and social functioning (as discussed by Cordier et al. [31]). Numerous competing definitions and theoretical approaches to social skills models exist [32,33,34,35]. In our review, we adopted a broad definition of social skills, considering them as behaviors performed in a social context involving interpersonal engagement. This aligns with Cordier et al. [31] and Wolstencroft et al.’s [7] perspectives.

## 2. Method

### 2.1. Search Strategy

A systematic literature search was conducted using PubMed and EBSCOhost (APA PsycINFO, Academic Search Complete, ERIC) databases from 2011 to October 2021. An update was conducted from the year 2021 to December 2023. The review was conducted according to PRISMA guidelines [36]. The following keywords were used to conduct the search: Autism OR Autism Spectrum Disorder OR ASD AND Serious Games OR Video Games OR Digital Games AND Social Skills Training OR Social Skills Intervention OR Social Skills Development. Filters for language (English, Portuguese, and French) were applied.

### 2.2. Eligibility Criteria

Employing the PICO (Population, Interventions, Comparison, Outcomes) framework [37], we established inclusion and exclusion criteria guided by the following research question: for children and adolescents with ASD, interventions using serious games, compared to other types of intervention or control conditions, are more effective in developing social skills? (cf. Table 1).

To be included, studies needed to meet the following criteria: (i) implement one or more interventions including at least one type of serious game; (ii) apply interventions directly to children and adolescents diagnosed with autism spectrum disorder, aged 0 to 18 years; (iii) use at least one quantitative assessment tool to evaluate social skills; and (iv) have a sample size equal to or greater than 15 participants, promoting in-depth discussion and yielding more precise results with diverse perspectives [38].

Exclusion criteria were defined as follows: (i) interventions only applied to parents, caregivers, or teachers of individuals diagnosed with ASD; (ii) studies lacking the application of a serious game; (iii) interventions targeting individuals diagnosed with other disorders, whether comorbid with autism spectrum disorder or not; (iv) studies lacking an assessment of social skills; (v) systematic reviews, meta-analyses, books or book chapters, and master’s or doctoral dissertations; and (vi) studies not published in English, Portuguese, or French.

### 2.3. Risk of Bias (Quality) Assessment

The Cochrane Collaboration’s RoB 2.0 tool [39] was applied to evaluate bias in randomized trials. This tool assesses five domains of bias focusing on different aspects of trial design, conduct, and reporting: (i) bias from randomization, (ii) bias from deviations in the intended intervention, (iii) bias from missing data, (iv) bias from outcome measurement, and (v) bias from the selection of the reported result.

To evaluate bias in non-randomized trials with a control group, we utilized the ROBINS-I [40], which assesses seven specific bias domains: (i) bias due to confounding, (ii) bias in the selection of participants for the study, (iii) bias in the classification of interventions, (iv) bias due to deviations from intended interventions, (v) bias due to missing data, (vi) bias in the measurement of outcomes, and (vii) bias in the selection of the reported result.

Two authors (A.C. and M.F.) independently assessed the risk of bias. Discrepancies were resolved through discussion. Since we anticipated a high risk of bias in most studies, we did not restrict analyses based on this parameter.

### 2.4. Study Selection, Data Extraction, and Management

The data extraction from all eligible articles was carried out in accordance with the PRISMA-P (Preferred Reporting Items for Systematic Review and Meta-Analysis Protocols [36]) diagram flow. Using the Rayyan software [41], we compiled the articles after obtaining the references. Initially, studies were identified based on title and abstract, adhering to the predefined inclusion criteria. Two researchers (A.C. and T.C.) independently evaluated the full search for inclusion, resolving discrepancies through discussion. A study was included if both reviewers independently deemed it met the inclusion criteria (Cohen’s κ = 0.98). In cases of disagreement, a third author (M.F.) mediated the decision. Extracted information included the year of publication, the age range of the participants, the number of participants in the intervention group and comparison groups, serious games used as an intervention, other interventions, duration/intensity of interventions, the target of the intervention, outcome measures, and main findings.

## 3. Results

### 3.1. Trial Flow

A total of 149 articles were identified through the specified search strategy across multiple databases. An additional 73 articles were added in an update. After eliminating 62 duplicates, 160 articles were screened based on their titles and abstracts. Of these, 135 reports were excluded for not meeting the inclusion criteria. Thus, 25 papers underwent a thorough full-text analysis, and 9 met the inclusion criteria. The trial flow is visually presented in Figure 1 through a PRISMA flow diagram.

### 3.2. Risk of Bias

Several authors incompletely reported design characteristics, thereby hindering an accurate assessment of the risk of bias. The assessment conducted by the two judges demonstrated almost perfect agreement (Cohen’s κ = 0.97). In summary, for the studies that included control groups, the review identified a high risk of bias across the randomized controlled trials (RCTs) included, with variations in high-risk areas observed among the studies (see Figure 2 for a table representing each study’s risk and Figure 3 for an overall summary of risk across the studies).

### 3.3. General Characteristics of the Included Studies

Table 2 presents a summary of the selected studies. Four studies were published between 2014 and 2016, while five were published between 2021 and 2023. The age distribution across the studies displayed variability, with the majority around 7 to 10-year-olds (eight studies). A smaller proportion of studies focused on children aged 5 (two studies) or above 12 (two studies). Notably, none of the studies included children below 5 years of age in their samples. The sample sizes also varied, with intervention groups ranging from 10 to 42 and control groups spanning from 10 to 40. This indicates a broad spectrum in the age and number of participants across the studies, highlighting the variability in the samples in the research under review.

Regarding the experimental design, this review included five randomized controlled trials (RCT) and four quasi-experimental design (QED) studies. The latter classification was utilized when participants were not randomly assigned to conditions or if additional details about the randomization process and potential biases in participant selection were unclear.

### 3.4. Intervention Characteristics

The duration of the interventions exhibited considerable variability, spanning from 4 to 40 weeks, with sessions taking place from 1 to 10 times a week. Additionally, each session ranged from 25 to 100 min, further emphasizing the diverse nature of the interventions across the studies.

The interventions implemented in the included studies targeted a spectrum of crucial domains, encompassing general social and communication skills, emotion regulation, and problematic behaviors. Specific interventions also addressed eye gaze, empathy, emotion recognition/encoding/decoding, joint attention, and symptoms associated with autism (i.e., dimensions including social impairments and circumscribed and repetitive behaviors and interests).

Various serious games have been utilized as interventions to enhance the social skills of individuals with autism: Emo Game software [42], Secret Agency Society (SAS; [51]), Social Games for Autistic Adolescents (SAGA; [44]), Zirkus Empathico (ZE; [52]), Immersive Virtual Reality System (IVRS; [46]), Play Emotion Detectives (ED; [53]), FaceSay program [54], JeStiMuLe [50], and Mind Reading [49]. These serious games are tailored to address distinct facets of social interaction, fostering the enhancement of social skills.

### 3.5. Serious Games Outcomes

The intervention outcomes were primarily assessed through questionnaires (e.g., Social Communication Questionnaire; [55], Social Skills Questionnaire; [56]; Emotion Regulation and Social Skills Questionnaire, [57]), although performance tests were also applied (e.g., NEPSY-II: Theory of Mind, NEPSY-II: Affect Recognition, [58]).

Results of each study were analyzed to assess the effectiveness of interventions on social skills, and 100% reported significant improvements in at least one of the constructs associated with social skills, as measured by performance assessment tools or questionnaires. Among the studies included in the analysis, seven directly examined the intervention’s impact on broad social skills [42,43,44,46,47,49,50]. Two studies specifically assessed emotional recognition abilities [45,47], and three focused on emotion recognition, encompassing facial and/or voice expressions [48,49,50]. Furthermore, one evaluated joint attention skills [48], two behavior skills [43,47], two eye gaze processing [44,48], one empathy [45], one autism symptomatology [49], and two explored emotion regulation [42,46]. This breakdown highlights the diverse range of outcomes assessed, showcasing the comprehensive nature of the studies in evaluating various aspects of intervention impact.

In these studies, significant effects were found across various domains: 85% demonstrated improvements in broad social skills, and 100% showed a positive impact on emotional recognition, perspective thinking, behavior problems, eye gaze processing, and empathy post-intervention. Additionally, all studies indicated a significant improvement in symptoms related to autism, and all reported enhancement in emotion regulation.

### 3.6. Characteristics of the Serious Games for Social Skills Training

The Emo Game computer games [42] employ a multimedia approach, adopting a simulation format. This game is strategically designed to enhance communication skills and emotional regulation. There are two distinct games: Emo Galaxy1 and Emo Galaxy2. In Emo Galaxy1, children should position their faces within the screen’s circle (as they have a camera) and choose a planet (each planet has an activity). After selecting the planet, the robot positioned at the center of the screen replicates and repeats the chosen activity. Successful completion of the activity by the child results in the background screen turning green, while an error during the task causes the screen to turn red. In Emo Galaxy2, children replicate a specified emotion and capture a photograph. Then, the background turns green for a suitable match or red for an error. Findings showed that the Emo Game effectively fosters emotional regulation among children diagnosed with ASD [42].

The Secret Agency Society (SAS; [43]) intervention was adapted from the SAS program [59]. It includes the SAS Computer Game, Visual Support Cards, parent training slides, and the Program Delivery Guide. The original version of this intervention endorses emotional and social skills. This game encourages the generalization of skills promoted in multiple contexts through the participation of family members and teachers. Research findings showed that individuals with ASD involved in this program demonstrated enhanced socioemotional functioning and problematic behaviors [43].

The Social Games for Autistic Adolescents (SAGA; [44]) aim to enhance sensitivity to eye gaze cues and social skills. The SAGA intervention helps adolescents with autism to understand eye gaze cues in simulated social interactions with computer-animated characters. Players learn to use eye gaze cues to guide their actions in the game, mirroring real-world developmental and practical applications. SAGA demonstrated efficacy in enhancing specific skills and improving real-life outcomes. This is particularly evident in the improved sensitivity to gaze signals, associated with improved social skills [44].

The Zirkus Empathico (ZE; [45]) is structured into four key sections: (i) fostering self-awareness of emotions; (ii) recognizing emotions portrayed in facial expressions; (iii) deducing emotions from contextual cues (cognitive empathy); and (iv) comprehending emotional resonance while simultaneously acquiring appropriate responses to others’ emotions. A supplementary fifth module includes an interactive animation, the emotion doll, integrated for real-life applications, explicitly targeting emotional communication within the family. Each module comprises multiple levels strategically designed to progressively increase in difficulty, contingent upon the child’s advancement in the preceding module. The training program is designed to be self-explanatory and employs a fox as a guiding character, offering instructions, explanations, and rewards throughout the process. It is strongly recommended that a caregiver supervise the training sessions to optimize emotional and empathetic communication skills development. Research findings suggested the efficacy of ZE in enhancing emotional awareness and regulation, concurrently mitigating general symptomatology associated with autism [45].

The Immersive Virtual Reality System (IVRS; [46]) is a virtual reality-based intervention that facilitates the identification and structuring of diverse social situations challenging for students, and teaching appropriate behaviors. In this intervention, children are immersed in a visual and interactive learning experience designed to foster the development of social skills. Through the simulation of various social scenarios or narratives, the IVRS guides students in acquiring suitable emotional responses, enhancing their emotional skills. Research findings underscored the IVRS as a valuable tool for acquiring and enhancing emotional skills among students diagnosed with ASD [46]. Research also suggested that immersive environments, in contrast to desktop virtual reality (VR) applications, significantly contribute to cultivating more appropriate emotional behaviors [46].

The Play Emotion Detectives (ED) game is grounded in the Crick and Dodges social information-processing model (1994, cited in Löytömäki et al. [47]). As such, children are expected to utilize their cognitive resources and past social experiences to interpret social-emotional cues within the game. The educational foundation of the game relies on the premise that decision-making and feedback within the gameplay foster novel experiences, paving the way for the adoption of new models of social behaviors. The game aims to accumulate fame points through “Office Tasks” and “Field Tasks”. The Office Tasks include 23 mini-games focusing on emotional learning domains, such as facial expression discrimination, emotional tone of voice recognition, and emotion vocabulary expansion. The Field Tasks incorporate metaphors and figurative language to enhance vocabulary learning. Research outcomes revealed that the ED game is a valuable tool for enhancing emotional discrimination and behavioral skills abilities for children with ASD [47].

The FaceSayTM [48] is a three-game program designed to teach specific facial processing skills to enhance social cognition. The first game focuses on identifying gaze direction and joint attention, the second on holistic facial recognition strategies, and the last on recognizing and identifying emotional expressions. Research suggests that this software enhances emotion recognition and comprehension, engaging in perspective-taking and social functioning of children with ASD [48].

The JeStiMulE [50] is a computer game that operates in a virtual reality setting, incorporating multisensory elements and interactivity through a gamepad. It features exercises designed to train emotion recognition in avatars, similar to conventional children’s video games. The JeStiMulE provides motivational instructions and allows players to create and navigate their avatars in a virtual environment, engaging in diverse real-life social situations. The primary goal of JeStiMulE is to enhance Theory of Mind skills, encompassing facial expressions, emotional gestures, and social scenarios. To achieve this, the game introduces nine expressions of basic emotions (e.g., happiness, anger, disgust, fear, sadness, surprise), a complex emotion (e.g., pain), and two complementary expressions (e.g., neutral and funny faces). Research results demonstrated the effectiveness of JeStiMulE in enhancing participants with autism ability to recognize emotions [50]. However, it is essential to note that the lack of a control group for comparison represents a limitation in the study design.

The Mind Reading (MR; [49]) is a computer-based intervention that enhances emotional stimulus decoding skills in individuals with high-functioning autism. The interactive software assists individuals in decoding facial expressions and prosody, employing visual and auditory lessons, practical tests, and computer-delivered reinforcements. More specifically, it helps children to recognize emotions through facial and vocal stimuli. It includes 98 selected emotions organized into emotional groups and levels. The research found that this tool is valuable for improving the encoding and decoding of emotions [49].

### 3.7. Characteristics of the Interventions in the Control Group

Eight studies compared the intervention group’s performance with active or passive control groups. In particular, four studies incorporated passive control groups employing a waitlist or business as usual. Additionally, four studies included active control groups, which involved groups receiving an alternative or active intervention for direct comparison with the experimental group undergoing the primary intervention: (i) the Central Intelligence Agency Condition; (ii) the SuccessMaker; (iii) other serious games focusing on non-social skills; and (iv) the Virtual Reality Software Application:

(i)The Central Intelligence Agency Condition (CIA) is a structured program incorporating the engagement of online cognitive activities with espionage themes (without social or emotional skills training components). The games are suitable for children aged 8 to 12 years old. Some web activities within the CIA include virtual jigsaw puzzles, finding differences between photos, and deciphering secret messages. The initial training and the weekly online coaching sessions for parents followed the same content and format as the SAS intervention parent sessions [43].(ii)The SuccessMaker is a computer-based course supplementing regular classroom reading instruction for grades K-8. Participants assigned to the control group underwent training sessions with specialized educators before the study. SuccessMaker aims to enhance reading comprehension through phonological awareness, phonetics, fluency, and vocabulary by utilizing lessons tailored to the participant’s reading level. Therefore, the overall goal of this intervention is to assist participants in developing and maintaining reading skills, providing opportunities for exploration, instruction, and the development of analytical abilities. Due to being an individualized program, it enables the participant to progress at their own pace. The computer analyzes the participant’s progress and introduces new skills according to what is deemed most appropriate [48].(iii)The serious games focusing on non-social skills aimed to promote children’s confidence in their actions/knowledge through computer-assisted parental guidance, intending to ensure a comparable level of parental interaction, motivation, and media use. Depending on the child’s age, various serious games were tailored to target non-social skills/knowledge, focusing on different content areas (e.g., traffic safety, body-related knowledge, school, nature). Similar to the ZE group, caregivers actively taught the training content to the children [45].(iv)The Virtual Reality Software Application (VRSA) is a software designed to compare learning outcomes with those facilitated by the IVRS. It was developed similarly to the IVRS but did not incorporate immersive environments [46].

## 4. Discussion

As individuals diagnosed with ASD face challenges in their daily lives, particularly in acquiring and developing social skills [30], the primary objective of this study was to systematically review the evidence on the application of serious games as a tool to enhance social skills in children and adolescents with this neurodevelopmental disorder. We explored the impact of serious games on distinct social skills and qualitatively analyzed the specific characteristics of these games. As expected, our findings showed that many of the outcomes of the studies included in this review found a positive influence of serious games on social skills and related domains, including emotion recognition/encoding/decoding regulation, eye gaze, joint attention, and behavioral skills. Nevertheless, caution is needed when interpreting the results due to the substantial risk of bias across various domains in the included studies.

The included studies display a wide variability regarding age distribution, sample sizes, and intervention duration/intensity. Furthermore, it is crucial to note the limited exploration of possible effects in children aged 5 and below and above 12, emphasizing a potential gap in the current body of research. Findings also highlighted the heterogeneous nature of serious game interventions for children with ASD, warranting further exploration and targeted investigation about specific characteristics of this type of game in specific age groups and intervention durations.

Within all the serious games analyzed in this review, the Secret Agency Society, Zirkus Empathico, and FaceSayTM are robust examples with a solid evidence-based foundation. This strength is primarily attributed to their evaluation through RCTs and the inclusion of active control groups. Incorporating RCTs in assessing these games significantly enhances the internal validity of the findings by establishing causal relationships and minimizing confounding variables. The random assignment of participants to either the intervention or control groups ensures a more accurate assessment of the games’ effects, enabling confident conclusions regarding their efficacy. Furthermore, including active control groups further increases their empirical strength, as researchers can compare these games to alternative interventions, helping to determine whether observed effects are specific to the games themselves or merely a result of non-specific factors, such as engagement.

One notable aspect of our findings is the role of serious games in addressing the specific challenges faced by individuals with ASD. Engaging with these children in face-to-face settings can often be difficult due to the lack of learner interest [1,2]. Therefore, the characteristics of serious games, such as the use of visual cues, the incorporation of dynamic and appealing characters, the opportunity to choose among various activities and environments, the opportunity for autonomy within the gaming environment, and the possibility for caregivers to be present, benefit these children and cater to this population’s unique needs and preferences [60]. Serious games offer a promising avenue for intervention in this population by providing a motivating and engaging platform for social skill development [13,28].

The results of the present review also highlight the use of immersive virtual reality technology. Lorenzo et al. [46] showed that more appropriate emotional behaviors were observed after immersive training than desktop virtual reality applications. Indeed, the use of immersive technologies can replicate scenarios and social meetings. Within these scenarios, the response and behavior are similar in virtual worlds as in the real world. This finding reflects and supports real-world conventions, understanding, and behaviors across diverse user groups [61,62,63].

## 5. Limitations and Implications for Practice and Future Research

Further research must increase the size of the participant’s sample and regular follow-ups to assess intervention effects, emphasizing the need for ongoing research to build upon these promising findings. Furthermore, we successfully identified RCTs assessing the efficacy of serious games, providing a robust methodological approach for evaluating the efficacy of interventions, establishing causation, and contributing to evidence-based practice. However, it is essential to acknowledge the limited scope of the available evidence. Systematically comparing the effectiveness of interventions in diverse social skill domains and examining the distinctions between digital and in-person approaches is crucial for achieving a comprehensive understanding of this topic. Using comparative analyses, researchers should discern whether serious games offer advantages, disadvantages, or comparable outcomes to other intervention methods. This type of comprehensive evaluation is essential for guiding future interventions, intervention planning, and decision-making regarding the most suitable approaches for addressing specific aspects of social skill development in children and adolescents with ASD. It ensures that interventions are effective and tailored to meet the unique needs of individuals within this population.

To enhance empirical evidence for implementing serious games as a complementary intervention for social skills for individuals with ASD in future studies, researchers should consider several key aspects:

(a) implement rigorous randomization procedures when allocating participants into the experimental and control groups, ensuring a fair and unbiased distribution of individuals across these groups;

(b) incorporate blinding strategies to minimize participants’ and informants’ expectations and mitigate potential placebo effects;

(c) compare the experimental group’s performance not only to a passive control group but also to an active control group. This comprehensive comparison helps elucidate the specific impact of social skills training compared to alternative interventions;

(d) utilize a combination of performance-based measures and ratings of social skills and behavior to assess the efficacy of the interventions. This dual assessment approach provides a more comprehensive understanding of the intervention’s impact on social skills development and real-world behavior;

(e) given the heterogeneity within the ASD population, characterized by varying cognitive, interpersonal, and physical skills, tailored interventions are imperative;

(f) explore the possible effects of the intervention, with serious games, on children aged 5 years and below and over 12 years old.

By addressing these considerations in future studies, researchers can contribute to a more robust and reliable body of evidence supporting the effectiveness of serious games as a complementary intervention for individuals with ASD.

Serious games have the potential to enhance adherence by fostering increased motivation and participation and constitute a unique avenue to encourage collaboration among individuals with ASD. While participative and collaborative games show promise for this population, concerns could also arise regarding the potential for increased isolation and other associated risks. Further research should consider these concerns in designing and implementing serious game interventions to ensure safe clinical practice. Like any tool, serious games can serve therapeutic or harmful purposes depending on their application and handling.

## 6. Conclusions

In summary, our review reveals a positive influence of serious games on social skills and related domains in children and adolescents. These include emotion regulation, recognition/encoding/decoding, eye gaze, joint attention, and behavioral skills. The promising outcomes suggest that serious games serve as effective interventions to enhance social skills. However, it is crucial to approach these findings cautiously due to the limited evidence available, highlighting the need for rigorous study designs to consolidate and strengthen these results. The limited evidence available emphasizes the necessity for robust study designs to consolidate results and integrate evidence-based intervention strategies.

Despite the current limitations, the positive trend warrants further investigation and justifies allocating resources to evaluate the efficacy of serious games as interventions. This calls for concerted efforts and resource allocation towards evaluating and refining these innovative approaches. The potential of serious games to positively influence social skill development holds profound implications for enhancing academic, interpersonal, and occupational outcomes in individuals with neurodevelopmental disorders. As such, continued research in this area is crucial for measuring the full potential of serious games as a valuable tool in promoting social competence and overall well-being.

## Figures and Tables

**Figure 1 healthcare-12-00508-f001:**
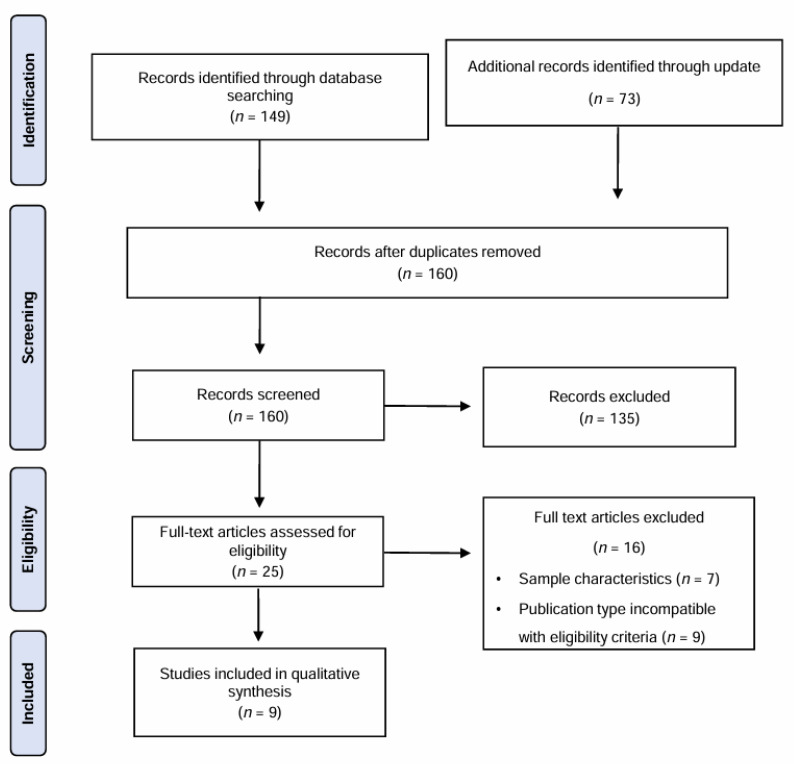
Preferred Reporting Items in Systematic Reviews and Meta-Analyses (PRISMA) flow diagram of selection of studies.

**Figure 2 healthcare-12-00508-f002:**
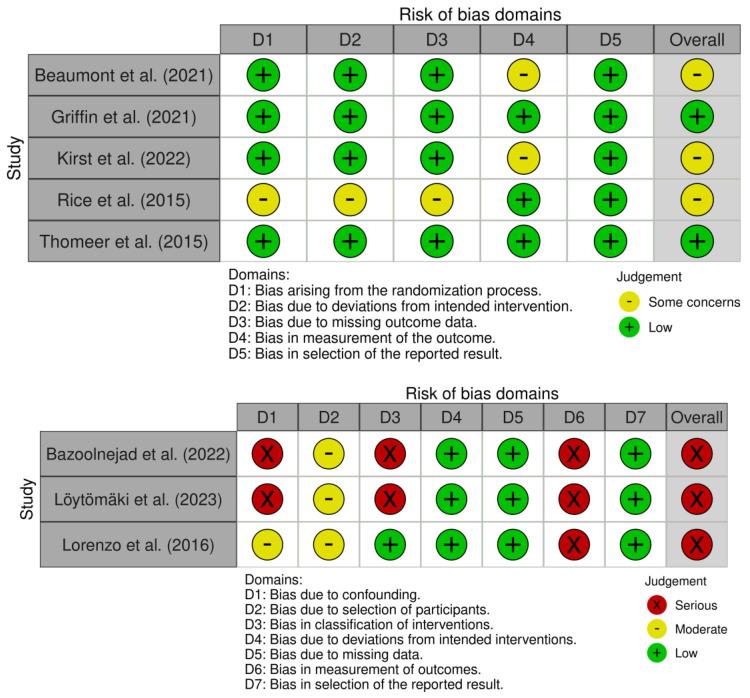
Risk of bias graph across included studies [42,43,44,45,46,47,48,49].

**Figure 3 healthcare-12-00508-f003:**
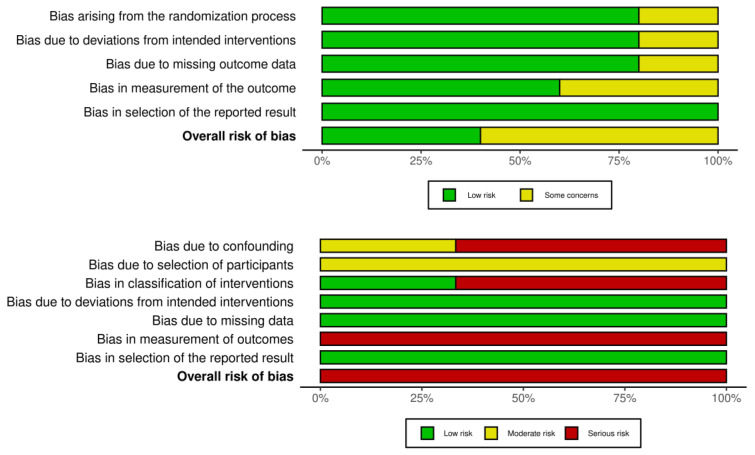
Risk of bias summary across included studies.

**Table 1 healthcare-12-00508-t001:** PICO (population/participant, intervention/indicator, comparator/control, outcome) framework.

PICO Framework	
Population	Children and adolescents with autism spectrum disorder
Intervention	Serious Games
Comparison	Other types of intervention or control conditions (when applicable)
Outcome	At least one domain of social skills

**Table 2 healthcare-12-00508-t002:** Characteristics of the included studies in the systematic review.

References (Year)	Method	Age Range	*N* Participants	Serious Game (Intervention)	Control Group	Duration	Targets of Intervention	Outcome Measures	Main Findings
Bazoolnejad et al. [42]	QED *	7–9	Intervention: 10 Control Group: 10	Emo Game	Waiting List	3 × 45 min per week (12 sessions)	Emotion regulation and communication skills.	- Social Communication Questionnaire; - Emotion Regulation Checklist.	After the intervention, results showed a significant difference between the experimental and control groups concerning emotion regulation. However, there was no significant difference in communication skills between the experimental and control groups.
Beaumont et al. [43]	RCT	7–12	Intervention: 35 Control Group: 35	Secret Agency Society (SAS)	Central Intelligence Agency Condition (CIA)	1 × 30 min per week (10 weeks)	Social skills and problematic behaviors.	- Social Skills Questionnaire; - Emotion Regulation and Social Skills Questionnaire; - Spence Children’s Anxiety Scale-Parent; - Eyberg Child Behavior Inventory-Parent.	After the intervention, participants in the experimental group showed more significant improvements than those in the control group in parent-rated social skills, problematic behaviors, and teacher-rated social skills.
Griffin et al. [44]	RCT	10–18	Intervention: 20 Control Group: 20	Social Games for Autistic Adolescents (SAGA)	Business as Usual	3 × 30 min per week (10 weeks)	Eye gaze processing and social skills.	- Gaze Perception Task; - Social Skills Improvement System; - Social Responsiveness Scale, Second Edition.	An interaction between group and time revealed that the experimental group demonstrated an increasing sensitivity to human eye gaze cues, while the business-as-usual group did not exhibit a similar progression. Despite being trained by computerized avatars to utilize eye gaze cues in the game, participants showed enhanced processing of human eye gaze cues in generalized learning tasks. Additionally, those in the intervention group who engaged in eye gaze tasks for at least 10 h displayed the most significant improvements in eye gaze sensitivity. This study also indicated that these enhancements correlated with heightened social communication skills, as reported by parents.
Kirst et al. [45]	RCT	5–10	Intervention: 42 Control Group: 40	*Zirkus Empathico* (ZE)	Other serious games focusing on non-social skills	At least 100 min per week, divided into two sessions (6 weeks)	Empathy, emotional states, and emotion recognition.	- Griffith Empathy Measure; - Kids Emotion Recognition Multiple Images Task; - Level of Emotional Awareness Scale for Children; - Emotion Regulation Checklist parent questionnaire; - Inventory of Callous-Unemotional Traits; - Social Responsiveness Scale; - Kiddy Kindl parent questionnaire; - Goal Attainment Scaling; - The Pediatric Volitional Questionnaire.	Training effects for empathy and emotion recognition were found post-intervention, although not sustained during follow-up. The short and mid-term assessments revealed moderate effects on emotional awareness, emotion regulation, and autism social symptomatology. Parents reported the attainment of treatment goals and positive training transfer.
Lorenzo et al. [46]	QED *	7–12	Intervention: 20 Control Group: 20	Immersive Virtual Reality System (IVRS)	Virtual Reality Software Application (VRSA)	4 × 35 min per month (40 weeks)	Emotional skills, emotional regulation competence during a change in a social situation.	- Emotional Script as a social script or behavior guideline in which the authors introduce ten social situations; - Computer vision (capable of automatically determining the child’s expression while the social stories take place); - Interviews with the students’ teachers.	Following the intervention period, the results highlighted a significant prevalence of more appropriate emotional behaviors in immersive environments as opposed to the utilization of desktop VR applications.
Löytömäki et al. [47]	QED	6–10	Intervention: 30 Control Group: 20	Play Emotion Detectives (ED)	Typically developing peers engaged in “Business as usual”	1 h at a minimum and 2 h at a maximum per week (eight weeks)	Emotion discrimination, emotion recognition, behavioral skills, social skills.	Six emotion discrimination tasks (Frankfurt Test and Training of Facial Affect Recogition; Nonsense words; Meaningful sentences; Photographs; Video clips; Matching task) and two questionnaires addressed to parents (Visual Analogue Scale and Strengths and Difficulties Questionnaire).	After the intervention, participants in the intervention group demonstrated significant improvements in their emotion discrimination skills across four tasks. In contrast, the control group exhibited significant progress in only one task, indicating limited improvement without regular practice. The gains observed in the intervention group were maintained at the one-month follow-up. According to parent reports, children in the intervention group showed some enhancements in emotion recognition and behavioral skills compared to the control group.
Rice et al. [48]	RCT	5–11	Intervention: 16 Control Group: 15	FaceSay	SucessMaker	1 × 25 min per week (10 weeks)	Eye gaze, joint attention, and facial recognition skills.	- NEPSY-II: Theory of Mind; - NEPSY-II: Affect Recognition; - Social Responsiveness Scale, Second Edition.	The FaceSay intervention improved participants’ capacity to recognize and comprehend emotions, engage in perspective-taking, and demonstrate improved social skills. In contrast, those who participated in the control group did not show comparable improvements in these areas.
Serret et al. [50]	QED	6–17	Intervention: 33	JeStiMuLe	-	2 × 60 min per week (4 weeks)	Emotion recognition, including facial expressions, emotional gestures, and social situations.	Five emotion recognition tasks involving 2D visual stimuli and were categorized into two types: emotions exhibited by avatars and emotions portrayed by real-life characters.	After the intervention, participants were more accurate at recognizing emotions.
Thomeer et al. [49]	RCT	7–12	Intervention: 22 Control Group: 21	MindReading	Waiting List	2 × 90 min per week (12 weeks)	Emotion recognition in faces and voices, emotion decoding and encoding, autism symptoms, and broad social skills.	- Cambridge Mindreading Face-Voice Battery for Children; - Emotion Recognition and Display Survey; - Social Responsiveness Scale; - Behavior Assessment System for Children, Second Edition; - Parent Rating Scales.	Following the intervention, more improved encoding and decoding of emotions were observed in the intervention group compared to the control group. Additionally, there was a notable decrease in symptoms of ASD. However, no significant difference was observed in social skills between the intervention and control groups.

Notes: RCT: randomized controlled trial; QED: quasi experimental design; NEPSY-II: A Developmental Neuropsychological Assessment, Second Edition; *N*: number of participants. *: to confirm if this study is a randomized controlled trial, additional details about the randomization process and potential biases in participant selection would be needed.

## Data Availability

Not applicable.

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
