# Peer review of "Serious Games for Developing Social Skills in Children and Adolescents with Autism Spectrum Disorder: A Systematic Review"

_healthcare, 2024, doi:10.3390/healthcare12050508_

Round 1

Reviewer 1 Report

Comments and Suggestions for Authors

This is nice and solid work which relates to an interesting topic that refers to autism in particular. It is suggested that the discussion and conclusion are strengthened more if discussed more in parallel with other authors' past work 

Reviewer 2 Report

Comments and Suggestions for Authors

Thank you for the opportunity to review this manuscript. Several revisions are needed before the manuscript can be published. 

1.     I suggest the authors combine the first two paragraphs in the introduction. It seems the two paragraphs discuss ASD characteristics. 

2.     For the third paragraph in the introduction, the authors said, “The social communicative challenges linked to ASD underscore the critical importance of targeting social skills in educational interventions.” I suggest the authors elaborate more on the “interventions” before you continue saying “Despite the increasing number of empirical studies and reviews on interventions for children with ASD.” I believe it is beneficial for readers to know what interventions are currently available.  

3.     The authors said, “no established standard practice has emerged for this particular group.” What does “established standard practice” mean? What does it have to do with your article topic? Please be clear.

4.     The authors said, “within a range of evidence-based interventions for improving social skills (e.g., real-world practice, video-based modeling, peer-assisted interventions, and group training),” this information should be introduced when you discussed interventions (See point 2). 

5.     Now it comes to a major point “serious game.”  After reading through your entire manuscript, I did not find a definition or a description of serious game. You should define and introduce the term in the introduction and include relevant citations. 

6.     You mentioned there are “several reviews have investigated the use of serious games as an adjunct intervention for individuals with autism ASD.” Why do you believe it is necessary for you to conduct the review? Although you mentioned there is a gap, could you specify what this gap entails? Please provide further elaboration on the gap, drawing from insights in current review articles. For instance, what limitations are identified in these existing review articles?

Method

1.     For the Eligibility Criteria outlined in Table 1, your objective is to compare the control group and intervention group. I suggest you incorporate an additional inclusion criterion specifying the presence of a control group.

2.     For the Study selection, have you calculated the agreement and disagreement during the article screening process? Similarly, for data extraction, have you calculated the agreement and disagreement? This information is crucial and should be included. Failing to address this could be a significant issue for a literature review. If not included, it should be explicitly mentioned as a limitation.

Results

1.     For “Characteristics of the Serious Games for Social Skills Training,” this section is informative. You should have the same information for the activities of the control group. You named activities for the control group in lines 205 to 212. I do not think you should include this information here. There should be a section specifically dedicated to the control group intervention. In this way, readers can compare both interventions and activities for the control group. 

2.     For “Serious Games Effects,” since you are not discussing “effect size,” you should not say effects. Please reword it. You can use “outcomes” or “dependent variables.”

Discussion and implications: 

1.     In your purpose, you said “Moreover, this review seeks to integrate therapeutic and game development perspectives, fostering communication between clinicians and game developers. The ultimate goal is to streamline the selection process for serious games used in interventions.” I did not see you include this information. If you did not include therapeutic and game development perspectives, you should remove this description in your purpose. 

2.     In your discussion, it is important to also explore how the game is tailored to support children with ASD. Within your results, where you outlined the characteristics of the interventions, please elaborate on how these characteristics specifically contribute to supporting children with ASD. Highlight any distinctive features that set it apart from other games.

3.     You mentioned “Furthermore, it is crucial to note the limited exploration of possible effects in children aged 5 and below and above 12, emphasizing a potential gap in the current body of research.” This should be included in implications for future research. 

Comments on the Quality of English Language

Moderate editing of English language required

Reviewer 3 Report

Comments and Suggestions for Authors

The Authors presented a systematic review on the use of serious games to improve social skills in children and adolescents with autism. Nine studies were reviewed in the last decade according to the eligibility criteria. Findings were critically discussed and future directions were suggestd. I feel that the manuscript is intersting and relevant for the Journal. I recommend its publication upon some minor issues to be solved in a suitable revision.My points are listed below.

1. The Rationale of the current Review should be further justified. Novelty feature should be argued. What this paper adds to the existing literature should be highlighted.

2. The Implications of the findings for both research and clinical practice should be further emphasized in the Discussion Section.

3. The Conclusion should be enhanced.

Round 2

Reviewer 1 Report

Comments and Suggestions for Authors

The suggested corrections have been dealt with and the paper can be published. Well done!

Reviewer 2 Report

Comments and Suggestions for Authors

The authors have addressed my comments. The manuscript should be ready for publication. 

Reviewer 3 Report

Comments and Suggestions for Authors

The manuscript can be accepted for publication.